# Nucleation and growth of orbital ordering

Takuro Katsufuji [1,2 ✉], Tomomasa Kajita [1], Suguru Yano [1], Yumiko Katayama[3] & Kazunori Ueno [3]

The dynamics of the first-order phase transitions involving a large displacement of atoms, for example, a liquid-solid transition, is generally dominated by the nucleation of the ordered phase and the growth of the nuclei, where the interfacial energy between the two phases plays an important role. On the other hand, electronic phase transitions seldom exhibit such a nucleation-growth behavior, probably because two-phase coexistence is not dominated by only the interfacial energy in such phase transitions. In the present paper, we report that the dynamics of a phase transition associated with an ordering of $d$ orbitals in a vanadate exhibits a clear nucleation-growth behavior and that the interfacial energy between the orbital-ordered and -disordered phases dominated by the orbital-spin coupling can be experimentally obtained.

[1] Department of Physics, Waseda University, Tokyo 169-8555, Japan. [2] Kagami Memorial Research Institute for Materials Science and Technology, Waseda University, Tokyo 169-0051, Japan. [3] Department of Basic Science, University of Tokyo, Meguro, Tokyo 153-8902, Japan. ✉email: katsuf@waseda.jp

Orbitals are one of the degrees of freedom in solids that interact with each other and order at low temperatures, similarly to the spin degrees of freedom. With the coupling of other degrees of freedom, orbitals often dominate various properties of materials. For example, a strong coupling between orbital and spin degrees of freedom leads to various intriguing phenomena in applied magnetic field, for example, the colossal magnetoresistance in various manganites[1] and magnetic-field-induced phase transitions in various vanadates[2,3]. Although details of the ordered state in these compounds have been experimentally investigated quite thoroughly, the dynamics or the fluctuations of orbitals are less well understood, probably because there are not many experimental techniques to detect the dynamics of orbitals. Phase transitions dominated by orbital ordering are often of the first-order, but their dynamics has not been clarified so far.

One of the typical dynamics in the first-order phase transition is the nucleation and the growth of the nuclei. A characteristic time dependence of the transformed volume fraction resulting from such a nucleation-growth process has been observed in the crystallization of liquid metals (particularly metallic glasses)[4–9], amorphous materials[10–12], the crystallization of polymers[13], and Martensitic transformations with a large displacement of atoms[14,15]. For such a nucleation-growth behavior to occur, the dominant energy must be the interfacial energy between the two phases, and this is typically the case for the phase transition with interfaces between solid and liquid, where almost negligible elastic energy appears as a bulk property of the domains.

The first-order electronic phase transitions with two-phase coexistence in crystals (Mott transition, magnetic transition, charge ordering, etc.) have been studied in various compounds, particularly focusing on the spatial distribution of the two phases[16–20]. The dynamics of the two-phase coexistence has also been studied in several compounds[21–25], but they barely exhibit conventional nucleation-growth behavior. One of the possible reasons for the absence of nucleation-growth behavior is that the two-phase coexistence is dominated not only by the interfacial energy but more dominated by the bulk properties, particularly by the elastic energy of the domains, in such phase transitions.

Recently, the dynamics of the first-order phase transition has been studied for some organic conductors exhibiting charge ordering, and a nucleation-growth behavior has been suggested on the basis of the time dependence of several physical quantities from the supercooled state[26–29]. This motivated us to search for the orbital-ordered material whose phase transition is dominated by the nucleation growth behavior. Here, one way to avoid the disturbance in the nucleation-growth behavior is to use the materials exhibiting a phase transition without variants (twin structures) of the ordered phase that possibly hinder the growth of domains.

From this viewpoint, we chose $BaV_{10}O_{15}$[30–34], which exhibits a sharp first-order phase transition at ~130 K caused by orbital ordering. In this compound, V ions form a bilayer quasi-triangular lattice and a structural phase transition characterized by V trimerization occurs below 130 K (Fig. 1), resulting in anomalies in various properties. This phase transition is caused by the ordering of the V $t_{2g}$ states; namely, at each side of the V trimer, the $d_{xy}$, $d_{yz}$, or $d_{zx}$ orbital of the neighboring V ions forms a bond in a spin-singlet state[34]. One of the characteristics in this phase transition is that in terms of the crystal structure, it is a phase transition from orthorhombic (Cmce) to orthorhombic (Pbca) and thus, no twin structure (variant) appears in the ordered phase. For more information on this compound, see Supplementary Note 1.

## Results

### Time dependence of resistivity, magnetic susceptibility, and strain.
The transition temperature of $BaV_{10}O_{15}$ itself is so high

that rapid cooling in a conventional manner cannot yield a supercooled state. We found that crystals with reduced transition temperatures can be obtained by doping Ti into the V site in this compound. Figure 2a, c shows the temperature (T) dependences of the resistivity ($\rho$) and the magnetic susceptibility ($\chi$) for $BaV_{10t-x}Ti_xO_{15}$ (BVTO), respectively. As can be seen, the transition temperature to the orbital-ordered state decreases from 130 K for $x = 0$ to ~70 K for $x = 0.15$, and almost disappears for $x = 0.20$. Furthermore, we found that $\rho(T)$ and $\chi(T)$ for $x = 0.15$ with a rapid cooling (200 K/min for $\rho$ and 50 K/min for $\chi$) and slow cooling (2 K/min for $\rho$ and 5 K/min for $\chi$) behave differently, as shown in Fig. 2b, d. For example, after rapid cooling of the sample, the resistivity is about two orders of magnitude smaller than that after slow cooling at the same $T$, indicating that the sample is still in the high-$T$ (HT) phase (a supercooled state), but with increasing $T$ at the rate of 2 K/min, resistivity increases at ~70 K and merges with that for the slowly cooled sample, indicating a phase transition to the low-$T$ (LT) phase. With further increasing $T$, $\rho$ decreases at ~90 K, indicating a phase transition to the HT phase. Such a reentrant behavior is commonly observed when the temperature is increased from a supercooled state[28,35]. Note that the reentrant behavior is not observed but $\chi(T)$ behaves similarly both after slow cooling and rapid cooling for $x = 0.10$, as shown in Fig. 2d. For more detailed cooling-rate dependence, see Supplementary Note 2.

To study the dynamics of the phase transition, we measured the time (t) dependences of $\rho$ (along the b-axis), $\chi$ (along the a-axis), and strain $\Delta L/L$ (along the b-axis) after rapid cooling (30 K/min for $\rho$ and $\Delta L/L$ and 50 K/min for $\chi$) from $T > 100$ K to various temperatures for BVTO with $x = 0.15$, which exhibits a supercooled state and a reentrant behavior for their $T$ dependences. As shown in Fig. 3a–c, all three quantities exhibit a clear $t$ dependence; $\rho$, for example, hardly changes but remains at $\rho_{HT}$ corresponding to the value in the HT phase, for a while after rapid cooling, but starts increasing at a certain time and is saturated at $\rho_{LT}$, corresponding to the value in the LT phase. As a rough estimate of the transformation time from the HT to the LT phase, $\tau_{1/2}$ at which $\log\rho$–$\log\rho_{HT}$ becomes a half of $\log\rho_{LT}$–$\log\rho_{HT}$ is plotted as a function of $T$ in Fig. 3d by red circles. As can be seen, $\tau_{1/2}$ obtained from $\rho$ decreases with increasing $T$ up to ~70 K, but it takes a minimum and then increases with further increasing $T$. Note that the $t$ dependences of $\chi$ and $\Delta L/L$ produce similar time–temperature–transformation (TTT) curves of $\tau_{1/2}$ represented by black crosses for $\chi$ and blue open squares for $\Delta L/L$ in Fig. 3d, which was estimated as the time at which $\chi$ or $|\Delta L/L|$ increases by half the increase from the HT to the LT phase. Such TTT curves have been observed in various systems exhibiting a first-order phase transition[8–11,13,27,28], but have not been observed for the electronic phase transition in inorganic materials. Note that the value of the minimum $\tau_{1/2}$ at ~70 K (~100 s) is sufficiently long so that the low-$T$ phase does not appear during the rapid cooling with a rate of several tens K/min for $x = 0.15$. For $x = 0.10$ or less, it is probably much shorter and thus, the supercooled state does not appear even with the rapid cooling. As another example of the materials exhibiting a similar TTT curve, the $t$ dependence of $\Delta L/L$ for the Sr-doped sample, $Ba_{1-x}Sr_xV_{10}O_{15}$ with $x = 0.06$, is shown in the Supplementary Note 3.

### Analysis of the time dependence by the Kolmogorov-Johnson-Mehl-Avrami model.
The time dependence of the transformed volume in the nucleation-growth process has been discussed using the Kolmogorov–Johnson–Mehl–Avrami (KJMA) model[36–39], which takes account of nucleation, the growth of the nuclei, and the impingement of the grown nuclei (see Supplementary Discussion 1). In this model, the time dependence of the transformed

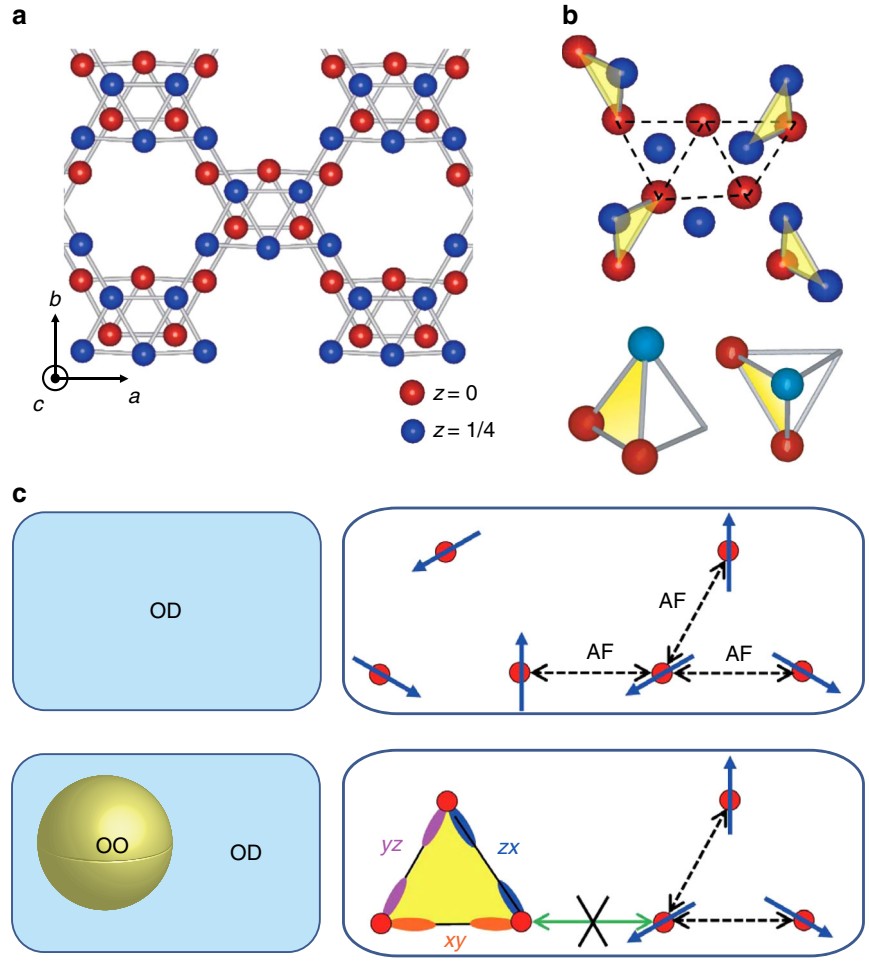

**Fig. 1 Crystal structure and the orbital and spin configurations for BaV$_{10}$O$_{15}$. a, b** Arrangement of the V ions **a** above and **b** below the transition temperature. In **b**, V trimers are highlighted in yellow. **c** (Left) Schematic of the orbital disordered (OD) phase (upper left) and the orbital-ordered (OO) cluster in the orbital disordered phase (lower left). (Right) Microscopic spin configurations and interactions in the OD phase (upper right) and at the boundary between the OO and the OD phases (lower right).

volume fraction $V(t)$ is given by

$$V(t) = 1 - \exp\left\{ -\left(\frac{t - \tau_0}{\tau}\right)^n \right\}, \qquad (1)$$

where $\tau$ is the transformation time and $n$ is the Avrami exponent, which is four in the case of isotropic three-dimensional growth but can be smaller if nucleus growth is anisotropic. $\tau_0$ (the transient time) comes from the time dependence of the nucleation rate $N(t)$. Namely, there is no nucleus of the low-$T$ phase immediately after rapid cooling to a certain temperature (i.e., $N(t = 0) = 0$), and it takes some time until the distribution of the size of various clusters attains a steady state and the time dependence of the nucleation rate $N(t)$ becomes constant value of $N_0$. The functional form of $N(t)$ has been calculated theoretically[40], but for the analysis of the experimental data, we assume that $N(t) = 0$ for $t < \tau_0$ and $N(t) = N_0$ for $t \geq \tau_0$.

If we assume $\tau_0 = 0$ in Eq. (1), the exponent $n$ and the transformation time $\tau$ can be obtained by plotting $\ln(-\ln(1-V(t)))$ as a function of $\ln t$[39]. Among the three quantities measured above, strain $\Delta L/L$ directly corresponds to the volume fraction of transformation, $V(t)$. Namely, the direction of the $a$-, $b$-, and $c$-axes in the orthorhombic structure does not change with the phase transition in the present compound and as a result, $\Delta L/L$ along any directions is proportional to $V(t)$ (see Supplementary

Note 4). Thus, we plot $\ln(-\ln(1-x(t)))$ with $x(t) = |\Delta L(t)/\Delta L(\infty)|$ as a function of $\ln t$ as shown in Fig. 4a, b. As can be seen, a clear kink structure, a subsequent linear increase, and another kink with saturation are observed in this plot above 76 K [Fig. 4a]. However, the slope, which corresponds to the exponent $n$, is larger than four above 78 K, which is an unphysical value. This result suggests that the transient time $\tau_0$ is nonzero above 78 K. We have fitted the data $x(t) = |\Delta L(t)/\Delta L(\infty)|$ with Eq. (1) with a finite $\tau_0$ (see Supplementary Note 5), and found that $\tau_0$ obtained by the fitting is positive for $T \geq 78$ K, but for $T \leq 77$ K, it is negligible or negative and is thus assumed to be 0 in the following. We plot $\ln(-\ln(1 - x(t)))$ as a function of $\ln(t - \tau_0)$ in Fig. 4c. As can be seen, all the data above 70 K exhibit a linear increase with a similar slope, $n = 3$–4.

On the hand, at lower temperatures ($T \leq 69$ K shown in Fig. 4b), the clear kink before the linear increase with $t$ is smeared out but a gradual increase appears for small $t$ in $\ln(-\ln(1 - x(t)))$, which is followed by an increase with a larger slope. It was found that the slope for the initial increase in the plot of $\ln(x(t))$ vs. $\ln t$ is close to 1/2 (see Supplementary Note 6), suggesting that the initial increase is dominated by one-dimensional diffusive motion of the interface[22]. Here, we assume that a transformation of a part of the region in the crystal is dominated by diffusive motion, and thus, $V(t)$ is given by the sum of the nucleation-growth formula

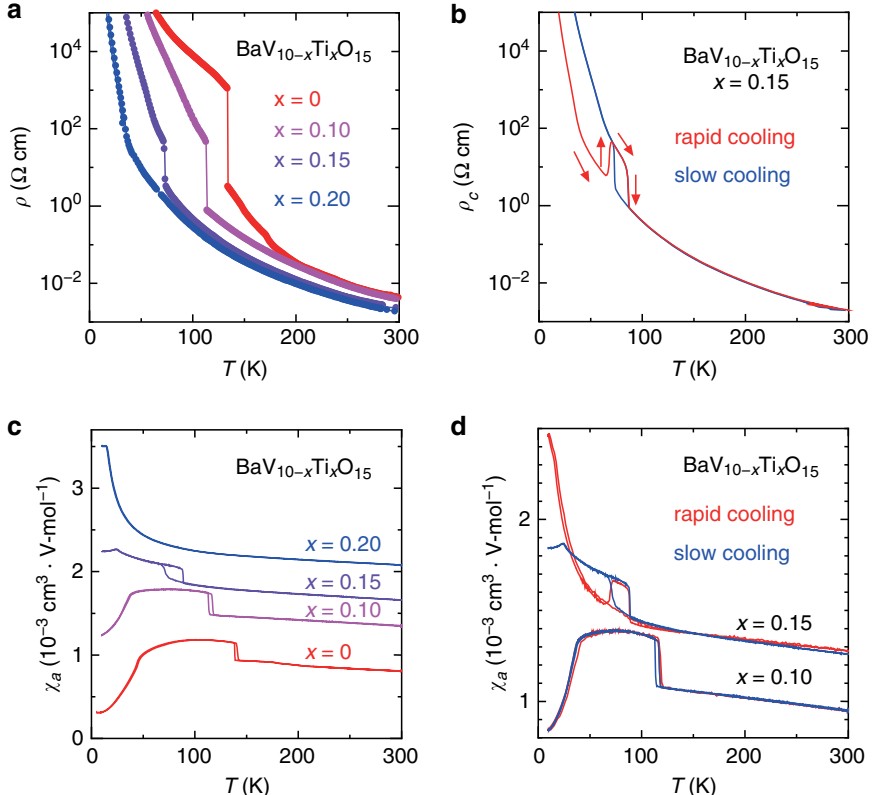

**Fig. 2 Temperature ($T$) dependence of resistivity and magnetic susceptibility for BaV$_{10-x}$Ti$_x$O$_{15}$ (BVTO).** **a** $T$ dependence of resistivity ($\rho$) for various values of $x$. **b** $T$ dependence of resistivity along the $c$ axis ($\rho_c$) for BVTO with $x = 0.15$ in a warming run (2 K/min) after rapid cooling (200 K/min, red) and in a cooling run and a warming run with a slow temperature change (2 K/min, blue). **c** $T$ dependence of magnetic susceptibility along the $a$-axis ($\chi_a$) for BVTO. **d** $T$ dependence of $\chi_a$ for BVTO with $x = 0.10$ and $x = 0.15$ with rapid cooling (50 K/min, red) and slow cooling (5 K/min, blue) measured both in a cooling run and a warming run (5 K/min).

given by Eq. (1) and that of the diffusive motion given by

$$V_{\text{diff}}(t) = \beta t^{1/2}, \qquad (2)$$

and fits the data obtained at $T \leq 77$ K. An example of the fitting result is shown by the dashed line in Fig. 4b for the data obtained at 55 K. This one-dimensional diffusive motion is probably caused by the anisotropic growth of domains dominated by the bulk elastic energy.

The transformation time $\tau$, the Avrami exponent $n$, the transient time $\tau_0$, and $\beta\tau^{1/2}$ (corresponding to the ratio of the region dominated by $V_{\text{diff}}$ to that dominated by the nucleation-growth behavior at $t = \tau$) obtained by the fitting above are shown in Fig. 4d–f. As can be seen, $\tau$ exhibits a minimum at ~70 K, $n$ is 3–4, $\tau_0$ decreases with decreasing $T$, and $\beta\tau^{1/2}$ increases with decreasing $T$ but is less than 0.15. These experimental results indicate that the dynamics of the phase transition is mainly dominated by the three-dimensional nucleation and growth with a finite transient time near the transition temperature. For additional data for the time dependence of various quantities, see Supplementary Note 7.

**Temperature dependence of the dynamics and interfacial energy.** Let us now discuss the $T$ dependence of the phenomena. According to the nucleation-growth theory (see Supplementary Discussion 2), the transformation time $\tau$ at each $T$ is proportional to $\left(N_0 v_0^3\right)^{-1/4}$, where $v_0$ is the constant speed for the interface and $N_0$ is the constant nucleation rate for $t > \tau_0$. The $T$ dependence of

$\tau$ dominated by $N_0$ and $v_0$ is given by[9,10]

$$\tau = \tau_{\text{ph}}\exp\left(\frac{T_1}{T}\right)\exp\left(\frac{T_2^3}{(T_c - T)^2 T}\right). \qquad (3)$$

Here, $T_1$ corresponds to the height of the energy barrier for adding one atom to the ordered phase, $T_2$ corresponds to the height of the energy barrier to overcome when the size of a cluster for the ordered phase exceeds a critical value, which is dominated by the interfacial energy between the two phases, and $\tau_{\text{ph}}$ is the inverse frequency of the characteristic phonon for the phase transition.

We fitted the $T$ dependence of the transformation time $\tau$ obtained experimentally from the time-dependent strain (shown by the solid circles in Fig. 4d) with Eq. (3) (shown by the solid line in the same figure), and obtained the following parameters; $T_1 = 2133$ K, $T_2 = 87.6$ K, $T_c = 103$ K, and $\tau_{\text{ph}} = 1.0 \times 10^{-15}$ s (see also Supplementary Note 8).

$T_2$ approximately corresponds to the interfacial energy per atom. Note that the interfacial energy between the A and B phases means that the energy is higher when the A and B components are next to each other than that when A and A or B and B are next to each other. For example, the interfacial energy between the two ordered states with different directions of the order parameter (in ferroelectrics or ferromagnets) is given by $\frac{K}{2}(\nabla m)^2$, where $m$ is the order parameter, by the Ginzburg–Landau theory, and $\nabla m$ has a finite value only at the interface. On the other hand, the interfacial energy between the ordered state and the disordered state in this picture is small since, in conventional phase transitions, it is unlikely that the energy cost between a

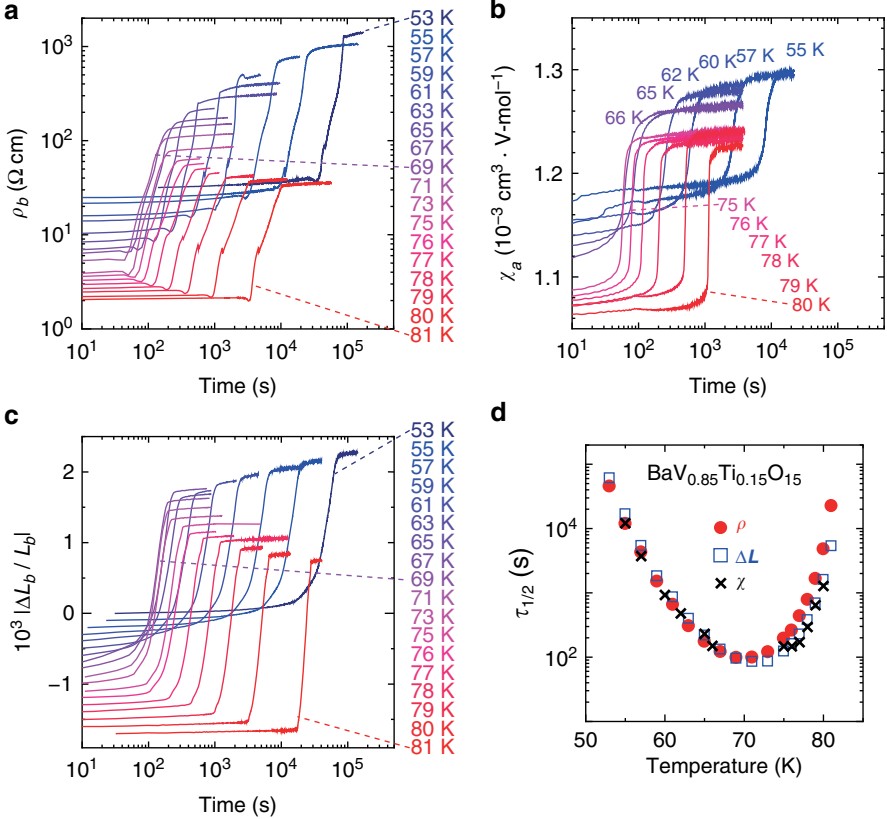

**Fig. 3 Time dependence of resistivity, magnetic susceptibility, and strain for BaV$_{10-x}$Ti$_x$O$_{15}$ with $x = 0.15$. a–c** Time dependence of **a** resistivity along the $b$ axis ($\rho_b$), **b** magnetic susceptibility along the $a$ axis ($\chi_a$), and **c** strain along the $b$ axis ($|\Delta L_b/L_b|$) after rapid cooling (30 K/min for $\rho$ and $\Delta L/L$ and 50 K/min for $\chi$) to each temperature. The data in **c** are offset for clarity. **d** $\tau_{1/2}$ estimated from $\rho_b$ (red circles), $|\Delta L_b/L_b|$ (blue open squares), and $\chi_a$ (black crosses). For the estimate of $\tau_{1/2}$ for each quantity, see the text.

component in the ordered state and that in the disordered state is larger than that between two components both in either the ordered or the disordered state. On the basis of the Ginzburg–Landau theory, such an interfacial energy corresponds to a four-body correlation function, which is smaller than two-body correlation functions, for example, in conventional systems with a single order parameter. Indeed, many of the nucleation-growth behaviors in the first-order phase transition are observed only in systems in which the phase transition is accompanied by a large displacement of atoms.

However, the situation is critically different in the present compound, where there are spin and orbital degrees of freedom that are strongly coupled with each other, leading to the Kugel–Khomskii interaction with a four-body term, $(s_i \cdot s_j)(\tau_i \tau_j)$ ($s_i$ and $\tau_i$ represent the spin and orbital degrees of freedom, respectively)[41]. In such a case, a change in the orbital state with orbital ordering can cause a change in the magnetic interaction, and accordingly, an energy cost can arise at the boundary between the orbital-ordered and -disordered states. In the case of the present compound, there is an $S = 1$ spin at each V site in the orbital-disordered phase, and these spins are antiferromagnetically (AF) coupled with each other (Fig. 1c, upper panels). In the orbital-ordered phase, on the other hand, the V spins forming trimers are in a spin-singlet state. Thus, if the orbital-ordered and -disordered phases are next to each other, the V spins in the orbital-disordered phase on the boundary cost energy since there are fewer spins to be AF coupled in the orbital-ordered phase. This leads to an interfacial energy dominated by the AF interaction (Fig. 1c, lower panels). Note that the change in the magnetic interaction with orbital ordering (larger AF coupling

within the V trimer and smaller coupling between the trimers) is essential to this interfacial energy. This picture is also consistent with the isotropic growth of nuclei experimentally observed, since, on the basis of the crystal structure, the magnetic interaction and the number of neighboring sites are fairly isotropic in the present compound.

We point out that the interfacial energy between the ordered and the disordered phases can affect the phase transition and thus will be important in general even if it does not exhibit the dynamics of the nucleation-growth behavior. One of the accomplishments in the present study is a quantitative estimate of such an interfacial energy from the experiment, $T_2 = 87.6$ K. The AF interaction $J$ of the present compound estimated from the Weiss temperature of the magnetic susceptibility ($\theta = s(s+1)zJ/3 = 1.2 \times 10^3$ K[33] with the number of nearest neighbor sites $z \sim 6$), is $\sim 300$ K, which is comparable to $T_2 = 87.6$ K, considering various factors omitted in the estimation of the interfacial energy.

Such orbital–spin coupling is an innovative mechanism of the interfacial energy, yielding a liquid-like domains with the electronic phase transition. Since such domains can be relatively easily controlled by temperatures and various external fields, they not only are the good systems to study the dynamics of the phase transition, but also can be a platform for intriguing physical properties arising from domains and interfaces.

## Methods

**Sample preparation.** The single crystals of BaV$_{10-x}$Ti$_x$O$_{15}$ were grown by the floating-zone technique. BaCO$_3$, V$_2$O$_3$, and Ti were mixed and pressed into a rod and was sintered at 1300 °C for 12 h in a flow of H$_2 = 7\%$/Ar gas. Then, the rod was melted by the floating-zone furnace with four halogen lamps in a flow of

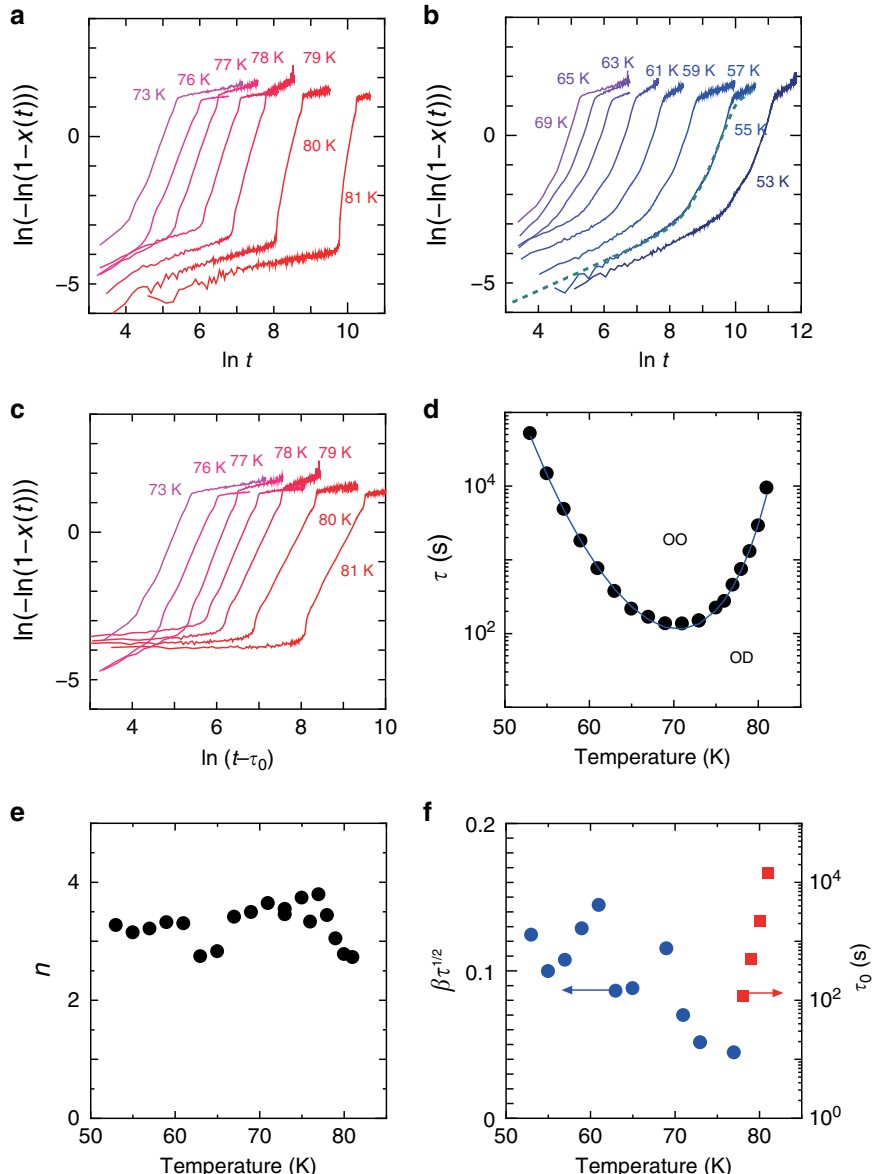

**Fig. 4 Analyses of the time dependence in the strain. a, b** $\ln(-\ln(1-x(t)))$ vs. $\ln t$, where $x(t) = |\Delta L(t)/\Delta L(\infty)|$ ($\Delta L$ is the strain along the $b$ axis) **a** above 73 K and **b** below 69 K. The dashed line in **b** is the curve fitted by the sum of Eq. (1) and Eq. (2). **c** $\ln(-\ln(1-x(t)))$ vs. $\ln(t-\tau_0)$, where $\tau_0$ is obtained by fitting with Eq. (1). **d** Transformation time $\tau$, **e** Avrami exponent $n$, and **f** diffusion component $\beta\tau^{1/2}$ and the transient time $\tau_0$ as a function of $T$ obtained by the fitting with the sum of Eq. (1) and Eq. (2). The solid line in **d** is the fitting with Eq. (3).

$H_2 = 7\%$/Ar gas with a feed speed of 8 mm/h. The orientation of the grown single crystals was determined by the Laue method and the crystals were cut in a rectangular shape.

**Measurement of resistivity, magnetization, and strain**. Resistivity of the sample was measured by a four-probe technique with silver paste as electrodes. Magnetization was measured by a SQUID magnetometer. Strain was measured by a strain-gauge technique. For rapid cooling in the resistivity and strain measurement, a heater (a strain gauge with a resistance of ~120 Ω) was attached to the sample, and the temperature of the cryostat is kept at a specific temperature and then, an appropriate magnitude of electric power was applied to the heater so that the temperature of the sample locally increases above 100 K. Then, we turned off the electric power to the heater and waited until the value of the resistivity or the strain becomes stable. After that, we started measuring the time dependence. For rapid cooling in the magnetization measurement, we used the temperature controller originally attached to the magnetometer, by which the rate in the temperature sweep can be increased to 50 K/min. More detailed description about the measurement is seen in the Supplementary Methods 1.

**Data availability**

The data that support the findings of this study are available from the corresponding author upon reasonable request.

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

## Acknowledgements
We thank K. Mochizuki and Y. Tabe for fruitful discussions. This work was supported by JSPS KAKENHI Grant no. 19H01853 and JST CREST Grant no. JPMJCR15Q2.

## Author contributions
T. Katsufuji and T. Katjita planned the project. T. Kajita and S.Y. grew the single crystals. T. Kajita, S.Y., Y.K., and K.U. performed the measurement of resistivity and strain. T. Kajita and S.Y. performed the measurement of magnetization. T. Katsufji wrote the paper. All authors discussed the results and commented on the paper.

## Competing interests
The authors declare no competing interests.
