## [Peer Review File · Nature Communications]

Reviewers' comments:

Reviewer #1 (Remarks to the Author):

This article explores the nucleation and growth phenomena in the transition into an orbitally ordered phase. By following the dynamics of different physical properties following rapid cooling into the ordered phase, the dynamics are modelled following expectations for different situations.

While the article is suitable for Nature Communications, there are several points that should be clarified before it is ready for publication:

Why do the authors present several doping levels and then focus on a specific one? There did not seem to be a motivation for the choice and it was not directly discussed.

Is the volume directly correlated with a single dimension? Often at phase transitions, the lattice volume is not conserved. Depending on the direction of ΔL measurement, this might be misleading. Equation two is for volume, then transitions to ΔX . Is that assuming that $V=x^3$? It is not very clear. I would think that this should change the equation and even the power law exponent. Is the crystal oriented to measure ΔL along a specific axis?

Cooling rates are different for ρ and χ ? Does this mean this rate does not matter within the differences? It is also a little misleading since the difference is two orders of magnitude for ρ but only one order for χ . Also, the cooling rate for the $\Delta L/L$ was never defined.

The manuscript comments on the domain wall diffusion being initially one-dimensional and then transitioning to three-dimensional. How does this work in terms of the physics? Is this tied to the nature of the lattice symmetry.

I think that the order of equation one should be reversed as they now transition from a specific to a general case. Seems it should be in the other order.

The title is very confusing as orbitals do not grow. However, domains of orbital order can grow.

Reviewer #2 (Remarks to the Author):

In the manuscript "Nucleation and growth of orbitals" the authors investigate the dynamics of the phase transition of occurring in (doped) BaV₁₀₀15 (BVO). Such transition is associated to the ordering of d-orbitals of the Vs and is accompanied by a structural change.

The main goal of the manuscript is to illustrate the nucleation-growth mechanism associated to such first order transition. The authors measure experimentally different characteristic quantities, such as resistivity, magnetic susceptibility and strain, both as a function of temperature (T) and time (t). By analysing the data against simplified theoretical descriptions the authors infer the relevance of orbital and spin coupling in determining the character of the nucleation growth dynamics across the phase transition.

The manuscript is well organized, yet the writing style is not always clear.

The results seem physically sound and technically correct. The experimental measures discussed in

the text are clear and properly illustrated although I have found that the figures captions often provide a poor description of the figures content making it hard for the reader to connect with the main text.

The main problem tackled in this study is interesting. Nevertheless, I do not judge it sufficiently relevant, or the findings of this manuscript innovative enough, to be worth of publication in Nature Communication. Although the problem of the dynamics at an electronic phase transition can be of some relevance for the community, the specific issue of nucleation growth seems of rather limited interest.

In particular the mechanism for nucleation growth discussed in the manuscript, indeed vaguely, and associated to the orbital and spin coupling in BVO does not appear to be solid enough to justify the putative high relevance of this work.

The coupling between spin and orbital degrees of freedom is a main feature of strongly interacting regime. This is clear also from the Kugel-Komskii form of the Hamiltonian in the strong coupling regime of a multi-orbital correlated system, as partly discussed by the authors in the final part of the paper.

Yet, different systems can display different dynamics and behaviors, not always associated to structural change as in the BVO. That is because other additional ingredients might become relevant near the transition point.

Thus one is not in principle authorised to use such generic motivation in a particular system to claim the generality of a mechanism.

There are however additional sources of criticism:

The introduction of the paper is rather vague and weak. For instance the position of the problem is based on the assumption that electronic phase transitions display negligible nucleation growth, which is not at all true in general. In facts phase separation and nucleation near criticality in Mott transition has been found in some compounds. So the motivation of the work is far less relevant than what the authors suggest.

The analysis of the experimental results is based on the KJMA model which express the nucleation growth in a simplified model. The related discussion is unnecessary long. All the different simplifications (with which I may agree) could have been discussed from the beginning, simplifying the analysis of the experimental data.

Concerning this point I can not avoid noticing that in this model there are at least three parameters, which recalls to my mind the famous quote of Fermi, in turn quoting Von Neumann "with four parameter I can fit an elephant, with five I can make him wiggle his trunk". This is to say that although formally correct, the predictive and explication power of this type of analysis is rather weak.

This becomes visible in the discussion at the conclusion of the work, which in my opinion results confused and rather weak. There are no direct and solid conclusions which can be drawn from the analysis, other than "we have indications that THIS transition comes with a nucleation growth process". Concerning the reasons why this is happening, those are essentially nearly speculations. I am not saying these are wrong, but just that the strenght of the conclusions is definitively low.

For all these reasons, although I believe this work can be published in some form elsewhere, I am convinced that it does not meet any of the criteria for acceptance in Nature Communication.

Reviewer #3 (Remarks to the Author):

In the manuscript, Nucleation and growth of orbitals, Katsufuji et al propose a method for monitoring phase transitions associated with orbital ordering in bulk single crystal $\text{BaV}_{0.85}\text{Ti}_{0.15}\text{O}_{15}$. In this material, a small amount of Ti was doped into the system to allow experimental access to desired cooling/warming profiles while maintaining a semblance of the parent material's ordering behavior. A range of time dependent measurements for resistivity, magnetic susceptibility, and strain were taken around the transition temperature. From these measurements, a series of transition lifetimes were obtained. These lifetimes were then used to compare expected outcomes from simplified models related to phase domain transition and propagation while also estimating energy requirements to transition between orbitally ordered and orbitally disorder states. To my knowledge this particular approach is novel in the context of orbital order/disordering processes and would be a useful tool for future works looking to understand domain dynamics dominated by spin-orbit coupling. As such, I recommend publication after the authors address the below comments and questions satisfactorily.

1. How does the additional Ti disorder in the lattice impact the transition dynamics? I would expect some impact to wall propagation in the presence of possible pinning sites and/or phonon scatterers. How can this be taken into account in the present model and/or what is the path to its development?

2. Figure 3 shows resistivity vs time for several different temperatures. There appears to be a two step process as resistivity shifts from low to high. That is to say that the transition regions have 2 different slopes. Is there some significance to this kink? If compared to figure 4d, is there some extra piece of understanding we can gain from this? Can this be taken as the onset of the initial percolative path followed by a slower growth of those initial seed points? Since the values of figure 4d are taken as the $\frac{1}{2}$ way point between the low and high resistivities, this difference in transition slope is likely skewing the results shown in figure 4d. It may be worth discussing this point.

3. The authors make the statement that TTT curves have been observed in various systems exhibiting first order phase transitions but not for electronic phase transitions. This statement could be misleading. Resistivity has been used to probe phase transition in the time domain. See the paper "Dynamics of a first-order electronic phase transition in manganites" (PRB 83, 125125) and "First-order reversal curve measurements of the metal-insulator transition in VO_2 " (PRB 79, 235110) These works and related confinement studies looking at phase transition dynamics in phase separated materials may also be useful in understanding the proposed percolative transport that muddies the resistivity measurements in the present manuscript.

Authors' Reply to Reviewer #1

We appreciate Reviewer #1 for evaluating our paper as “suitable for Nature Communications”. We also thank the reviewer’s various insightful comments, all of which were very important to improve the quality of the present paper. We respond to each comment below:

Comment 1: Why do the authors present several doping levels and then focus on a specific one? There did not seem to be a motivation for the choice and it was not directly discussed.

Response: As shown in Fig. 2d, $\text{BaV}_{10-x}\text{Ti}_x\text{O}_{15}$ (BVTO) with $x=0.10$ does not exhibit a supercooling behavior, but the magnetic susceptibility behaves similarly both with rapid cooling and slow cooling. Thus, we cannot obtain the lower- T side of the TTT curve for this sample, since a supercooled state is necessary to measure the time dependence in this region. The reason why the supercooled state was not obtained for $x=0.10$ is that the transformation time τ at the minimum region in the TTT curve is probably shorter than that for $x=0.15$, in which τ at the minimum is $\sim 10^2$ s. This issue was not discussed properly in the old version of the paper, and in the revised version, we discuss this issue more explicitly (“Note that the reentrant behavior is not observed but...” in p3, “which exhibits a supercooled state and a reentrant behavior...” in p4, and “Note that the value of the minimum t at ~ 70 K (~ 100 s) is sufficiently long...” in p5.) On the other hand, BVTO with $x=0.20$ does not exhibit a phase transition. Thus, we can perform the measurement of the time dependence only for $x=0.15$ in the series of BVTO.

To support the present experimental result, we show the time dependence of the strain for $\text{Ba}_{1-x}\text{Sr}_x\text{V}_{10}\text{O}_{15}$ (BSVO) with $x=0.06$ newly in the revised supplement material (“Time dependence of the strain for $\text{Ba}_{1-x}\text{Sr}_x\text{V}_{10}\text{O}_{15}$ ”). The similarity between the result for BVTO and that for BSVO confirms that this behavior is an intrinsic one in the present systems. The reason why we use BVTO for the analysis in the present paper is that (1) the data are missing between 58 and 75 K for BVSO because of too short transformation time in this T range and (2) inhomogeneity is larger for BVSO judging from the experiment results on the x dependence of various properties.

Comment 2: Is the volume directly correlated with a single dimension? Often at phase

transitions, the lattice volume is not conserved. Depending on the direction of ΔL measurement, this might be misleading. Equation two is for volume, then transitions to ΔX . Is that assuming that $V=x^3$? It is not very clear. I would think that this should change the equation and even the power law exponent. Is the crystal oriented to measure ΔL along a specific axis?

Response: First, we used a single crystal and we measured the strain along the b axis. It was not properly described in the old version of the paper, and in the revised manuscript, we explicitly describe this (“To study the dynamics of the phase transition, we measured...” in p3.) Furthermore, since the phase transition is from orthorhombic to orthorhombic in terms of the crystal structure, the relationship between a , b , and c axes does not change with the phase transition. This is also explicitly described in the revised version of the paper (“Namely, the direction of the a , b , and c axes in the orthorhombic structure does not change...” in p7.) Finally, the reason why the strain along a certain direction is proportional to the volume fraction is described newly in the supplemental material “Relation between the strain and the volume fraction in composite materials”. Roughly speaking, we consider the material composed of $N = N_1 \times N_2 \times N_3$ rectangular parallelepipeds either in the A or B phases. If the volume fraction of the A phase is x , then the number of the parallelepipeds in the A phase is xN . If we consider the change in the length in a certain direction along which there are, for example, N_1 parallelepipeds, the number of those in the A phase is also xN_1 . Thus, the strain is proportional to the volume fraction x .

Comment 3: Cooling rates are different for ρ and χ ? Does this mean this rate does not matter within the differences? It is also a little misleading since the difference is two orders of magnitude for ρ but only one order for χ . Also, the cooling rate for the $\Delta L/L$ was never defined.

Response: First, the cooling rate for $\Delta L/L$ was missing in the old version of the paper, and it is described in the revised version in p4 (30 K/min). Next, there is threshold in the cooling rate to achieve the supercooled state in the present system. As shown in the new supplemental material (“Magnetic susceptibility measured with various sweep rates of the temperature”), if cooling rate is higher than 20 K/min, the system can reach the supercooled state, whereas if it is lower than 5 K/min, the system completely goes into the low- T state. Thus, the issue is not the ratio of the cooling rates for the rapid cooling and the slow cooling, but whether it is above or below the threshold. This threshold is

related to the minimum of τ in the TTT curve, and this is also discussed in the revised version. (“Note that the value of the minimum t at ~ 70 K (~ 100 s) is sufficiently long...” in p5-6.)

Comment 4: The manuscript comments on the domain wall diffusion being initially one-dimensional and then transitioning to three-dimensional. How does this work in terms of the physics? Is this tied to the nature of the lattice symmetry.

Response: The diffusive motion given by Eq. (2) is correlated with the elastic properties of the domains. and since the change in the lattice constants with the phase transition is anisotropic (i.e., the b lattice constant decreases whereas the a and c lattice constants increase from the high- T to the low- T phase), it is possible that the growth of the domain dominated by the diffusive motion is much faster along a certain direction, resulting in the one-dimensional behavior. We have added this description in p9 (“This one-dimensional diffusive motion is probably...”). It should be emphasized that the phase transition occurring over 85 % of the sample is dominated by the three-dimensional nucleation-growth behavior given by Eq. (1) and that occurring only in 15 % is dominated by the one-dimensional diffusive motion given by Eq. (2).

Comment 5: I think that the order of equation one should be reversed as they now transition from a specific to a general case. Seems it should be in the other order.

Response: We appreciate this comment. We have changed the order of the explanation so that the functional form used for the fitting appears first as Eq. (1) (p6-7).

Comment 6: The title is very confusing as orbitals do not grow. However, domains of orbital order can grow.

Response: We appreciate this comment. We have changed the title to “Nucleation and growth of orbital ordering”.

Authors' Reply to Reviewer #2

We appreciate Reviewer #2 for evaluating our paper as “the manuscript is well organized” and “the results seem physically sound and technically correct.” On the other hand, Reviewer #2 claims that “I do not judge it sufficiently relevant or findings innovative enough to be worth of publication in Nature Communications”. We are afraid that Reviewer #2 underestimates the accomplishment of the present study on the “dynamics” of the phase transition, and it is partly because the introduction in the old version of the paper was not sufficient to clearly explain the issue of dynamics in the phase transition. Thus, we have quite thoroughly revised the introduction and have made the point clearer in the revised version.

Reviewer #2 also gave various insightful comments and we respond to them below:

Comment 1: Although the problem of the dynamics at an electronic phase transition can be of some relevance for the community, the specific issue of nucleation growth seems of rather limited interest.

Response: We respectfully but strongly disagree with this judgement of “limited interest”. The nucleation-growth behavior is one of the dynamics most commonly observed in various first-order phase transitions, though it has never been observed in the electronic phase transitions of inorganic compounds previously. As discussed in the introduction, such dynamics has recently been observed in the charge ordering of organic conductors, as reported in two *Science* papers (Ref. 20 and 21 in the present paper). We believe that the present paper is highly important in a sense that the electronic phase transition dominated by nucleation-growth behavior is discovered for the first time in a transition-metal oxide as another typical strongly correlated electron system.

Comment 2: In particular, the mechanism for nucleation growth discussed in the manuscript, indeed vaguely, and associated to the orbital and spin coupling in BVO does not appear to be solid enough to justify the putative high relevance of this work. The coupling between spin and orbital degrees of freedom is a main feature of strongly interacting regime. This is clear also from the Kugel-Komskii form of the Hamiltonian in the strong coupling regime of a multi-orbital correlated system, as partly discussed by the authors in the final part of the paper.

Response: As the reviewer pointed out, we claim that the Kugel-Khomskii interaction is the main source of the interfacial energy in the present compound. We have made a rough estimate of the magnitude of the interaction and compare it with the experimentally obtained interfacial energy in the second last paragraph of the paper. One of the difficulties in making more quantitative analysis is to theoretically handle the spin-singlet state in the V trimer, and it is our ongoing work. Thus, the theoretical analysis of the interfacial energy is beyond the scope of the present paper. Nevertheless, we believe that the issue (the interfacial energy by the Kugel-Khomskii interaction) is discussed clearly in the paper.

Comment 3: Yet, different systems can display different dynamics and behaviors, not always associated to structural change as in the BVO. That is because other additional ingredients might become relevant near the transition point. Thus one is not in principle authorised to use such generic motivation in a particular system to claim the generality of a mechanism.

Response: We may not understand this comment correctly, but we do not claim that a nucleation-growth model can be applied to any orbital ordering of transition-metal oxides. For example, many electronic phase transitions are dominated more by the elastic energy of the domains, rather than the interfacial energy between domains, and therefore, their dynamics is different from the nucleation-growth behavior. Nevertheless, we point out two aspects of generality in the present study: (1) The nucleation-growth behavior is one of the most common dynamics in the first-order phase transitions, though it has never been observed in the electronic phase transitions of inorganic compounds. We think that the discovery of the nucleation-growth behavior in those materials itself is important in this respect. (2) From the analysis, we quantitatively obtained the interfacial energy between the orbital-ordered and orbital-disordered phases. This energy is relevant even to the materials that do not exhibit a nucleation-growth behavior in the phase transition (because of the large elastic energy). In other words, such an interfacial energy may affect the phase transition in general. We have emphasized the issue of (2) in the revised version of the paper (p10-11. “We point out that the interfacial energy between...”)

This reviewer’s comment has also reminded us of the importance of discussing why the present compound exhibits a nucleation-growth behavior whereas others do not. In the introduction of the revised version, we have added the discussion on the absence of variants (twin structures) in the phase transition of the present compound, which leads to

reducing the unnecessary elastic energy of the domains (p3 “Here, one way to avoid the disturbance...” and “One of the characteristics in this phase transition...”).

We also point out that the statement “one is not in principle authorised to use such generic motivation in a particular system to claim the generality of a mechanism” seems to deny the research of finding new materials that exhibit intriguing physical properties. We know that different materials exhibit different behaviors, but we try to find a new material that exhibits a behavior involving intriguing physics that has never been observed before. In the present case, a new material ($\text{BaV}_{10}\text{O}_{15}$) exhibits a novel behavior (the dynamics of orbital ordering) involving intriguing physics (nucleation-growth behavior dominated by the orbital-spin interactions). We believe that it has sufficient generality as research.

Comment 4: the position of the problem is based on the assumption that electronic phase transitions display negligible nucleation growth, which is not at all true in general. In fact phase separation and nucleation near criticality in Mott transition has been found in some compounds. So the motivation of the work is far less relevant than what the authors suggest.

Response: We are afraid that there is a confusion here between “phase separation” and “the dynamics of the nucleation-growth behavior”. We know that many compounds exhibit phase separation associated with the phase transition, and the spatial distribution of domains have been studied by various techniques. On the other hand, there have not been so many studies on the dynamics of the phase transition, and none of the electronic phase transition in inorganic compounds exhibits a conventional nucleation-growth behavior. It should be emphasized that there are several different dynamics on the phase separated states, for example, nucleation-growth and spinodal decomposition, and the nucleation-growth dynamics involves the process where many “embryos” appear and only some of them exceed a critical size and become “nuclei” that keep increasing in size. In other words, we are discussing a peculiar type of dynamics, which cannot be investigated only by the techniques with spatial resolution.

To make this point clear, we have modified the introduction, where we cite several papers (Ref. [16]-[20] in the revised version) on the study of the spatial distribution of the coexisting two phases. (p2. “The first-order phase electronic phase transition...particularly focusing on the spatial distribution of the two phases...”)

Comment 5: The analysis of the experimental results is based on the KJMA model which express the nucleation growth in a simplified model. The related discussion is unnecessary long. All the different simplifications (with which I may agree) could have been discussed from the beginning, simplifying the analysis of the experimental data.

Response: We appreciate the reviewer's insightful comment and we agree to the suggestion. In the revised version, we show the functional form used for the fitting from the beginning [Eq. (1) and p6-7].

Comment 6: Concerning this point I cannot avoid noticing that in this model there are at least three parameters, which recalls to my mind the famous quote of Fermi, in turn quoting Von Neumann "with four parameter I can fit an elephant, with five I can make him wiggle his trunk". This is to say that although formally correct, the predictive and explication power of this type of analysis is rather weak.

Response: In the present study, (a) the time dependence of various quantities has been studied and analyzed based on Eq. (1). In particular, the exponent n is important to prove the applicability of the nucleation-growth model, and we found that n is close to 4, consistent with the three-dimensional nucleation-growth model. Furthermore, (b) the temperature dependence of the transformation time obtained in (a) has been analyzed based on Eq. (3). We are afraid that the referee only criticizes the analysis in (b) but underestimates the importance of (a). Furthermore, the purpose of the analysis in (b) is not to "prove" that our analysis is correct but to obtain the interfacial energy, corresponding to T_2 in Eq. (3), experimentally.

Comment 7: This becomes visible in the discussion at the conclusion of the work, which in my opinion results confused and rather weak. There are no direct and solid conclusions which can be drawn from the analysis, other than "we have indications that THIS transition comes with a nucleation growth process". Concerning the reasons why this is happening, those are essentially nearly speculations. I am not saying these are wrong, but just that the strength of the conclusions is definitively low.

Response: As discussed above, one of the main accomplishments in the present study is the quantitative estimate of the interfacial energy between the orbital-ordered and -disordered phases. This interfacial energy is a novel concept, and it should exist in any materials showing orbital ordering, though it has never been discussed and there has been

no quantitative estimate of such interfacial energy. We believe that this is one of the strong conclusions in the present study. We have emphasized this point in the revised version (p11, “One of the accomplishments in the present study is a quantitative estimate of the interfacial energy.”)

Furthermore, to make the conclusion stronger, we have added a description about the easiness of controlling the domain structure in this compound so that it can be a good platform to study the domains and interfaces. (p11 “Such orbital-spin coupling is an innovative mechanism...”)

Comment 8: I have found that the figures captions often provide a poor description of the figures content making it hard for the reader to connect with the main text.

Response: We have added descriptions in the figure captions so that they become more self-contained.

Authors' Reply to Reviewer #3

We appreciate Reviewer #3 for evaluating our paper as “novel in the context of orbital order/disordering processes” and “useful tool for future works looking to understand domain dynamics dominated by spin-orbit coupling”. We also thank the reviewer’s various insightful comments, all of which were very important to improve the quality of the present paper. We respond to each comment below:

Comment 1: How does the additional Ti disorder in the lattice impact the transition dynamics? I would expect some impact to wall propagation in the presence of possible pinning sites and/or phonon scatterers. How can this be taken into account in the present model and/or what is the path to its development?

Response: It is quite difficult to answer this question only from the experiment on $\text{BaV}_{10-x}\text{Ti}_x\text{O}_{15}$ (BVTO). Thus, we show the data of the time dependence of the strain for $\text{Ba}_{1-x}\text{Sr}_x\text{V}_{10}\text{O}_{15}$ (BSVO) with $x=0.06$ in the revised supplement material. It is likely that Ti, which is substituted for V with the orbital degree of freedom, has a larger impact in terms of the disorder on the domain wall compared with Sr, which is substituted for Ba away from V. Nevertheless the behaviors are not different so much between the two systems, indicating that the effect of the disorder on the wall propagation is not crucial to the present experimental results.

The reason why we use BVTO for the analysis in the present paper is that (1) the data are missing between 58 and 75 K for BVSO because of too short transformation time in this T range and (2) inhomogeneity is larger for BVSO judging from the experiment results on the x dependence of various properties.

Comment 2: Figure 3 shows resistivity vs time for several different temperatures. There appears to be a two step process as resistivity shifts from low to high. That is to say that the transition regions have 2 different slopes. Is there some significance to this kink? If compared to figure 4d, is there some extra piece of understanding we can gain from this? Can this be taken as the onset of the initial percolative path followed by a slower growth of those initial seed points? Since the values of figure 4d are taken as the $\frac{1}{2}$ way point between the low and high resistivities, this difference in transition slope is likely skewing the results shown in figure 4d. It may be worth discussing this point.

Response: This has been discussed already in the original version of the supplemental information in “Avrami plot of other quantities.” (and now in “Time dependence of other quantities”). The two-step process in the time dependence of the resistivity is likely caused by the inhomogeneity of the sample. This means that the strain measurement by the strain gauge is dominated mainly by the region near the strain gauge in the sample, but the resistivity measurement is dominated by the whole region of the sample and thus, is subject to inhomogeneity more. Note that only the variation of the transition temperature by several hundred millikelvin in the sample results in this kind of seemingly anomalous behavior.

In addition, the resistance measurement is not bulk sensitive but is dominated by how the electric current flows inside the sample. Since there is anisotropy in the resistivity for the present compound, inhomogeneity in the electric current density in the sample can lead to a complicated behavior of the resistance. We newly point out in the supplemental information that there is a small decrease in the resistivity before it starts to increase, as clearly seen in Fig. S7 (a). This may also come from such a complication with inhomogeneity and anisotropy.

Thus, we think that it is highly difficult to quantitatively analyze the time dependence of the resistivity shown in Fig. 3a. Nevertheless, we have added the analysis of the resistivity based on the effective media approximation in Fig. S7 (b), as discussed below.

Regarding the accuracy of Fig. 4d, since this is the transformation time obtained by the fitting of the strain by Eq. (1) as illustrated in Fig. 4b and Fig. S4, there is no problem arising from such an issue.

Comment 3: The authors make the statement that TTT curves have been observed in various systems exhibiting first order phase transitions but not for electronic phase transitions. This statement could be misleading. Resistivity has been used to probe phase transition in the time domain. See the paper “Dynamics of a first-order electronic phase transition in manganites”(PRB 83, 125125) and “First-order reversal curve measurements of the metal-insulator transition in VO₂”(PRB 79, 235110) These works and related confinement studies looking at phase transition dynamics in phase separated materials may also be useful in understanding the proposed percolative transport that muddies the resistivity measurements in the present manuscript.

Response: We appreciate the referee's useful input on the previous papers. We have cited the papers (Ref. [24][25]) properly in the revised version, and we have modified the description so that it does not give the impression to the readers that there has been no study on the dynamics of electronic phase transitions. (p2. "The first-order phase electronic phase transition...") Furthermore, inspired by the work in PRB 79, 235110, we have added in the supplemental material the data of the resistivity calculated by the effective media approximation (EPA) based on the volume fraction of the two phases estimated from the strain measurement (Fig. S7 (b)), though the result cannot perfectly explain the experimental result on the resistivity.

We emphasize that the advantage in the present study is to use the strain measurement to estimate the volume fraction of the two phases instead of the resistivity measurement. This becomes possible because of the bulk single crystal with a large size and good quality.

REVIEWERS' COMMENTS:

Reviewer #1 (Remarks to the Author):

This article explores the nucleation and growth phenomena in the transition into an orbitally ordered phase. By following the dynamics of different physical properties following rapid cooling into the ordered phase, the dynamics are modelled following expectations for different situations.

I feel that the authors have satisfactorily responded to my concerns and now the paper is ready for publication.

Reviewer #3 (Remarks to the Author):

All of my questions and suggestions have been addressed to my satisfaction.